# Monetising Air Pollution Benefits of Clean Energy Requires Locally Specific Information

**Mandana Mazaheri [1], Yvonne Scorgie [1], Richard A. Broome [2], Geoffrey G. Morgan [3] , Bin Jalaludin [4] and Matthew L. Riley [1,\*]**

1. New South Wales Department of Planning, Industry and Environment, Parramatta 2124, Australia; Mandana.Mazaheri@planning.nsw.gov.au (M.M.); yvonne.scorgie@environment.nsw.gov.au (Y.S.)
2. New South Wales Ministry of Health, Health Protection NSW, Sydney 2060, Australia; Richard.Broome@health.nsw.gov.au
3. University Centre for Rural Health, School of Public Health, University of Sydney, Lismore 2480, Australia; geoffrey.morgan@sydney.edu.au
4. School of Public Health and Community Medicine, University of New South Wales, Sydney 2006, Australia; b.jalaludin@unsw.edu.au
* Correspondence: matthew.riley@environment.nsw.gov.au

**Abstract:** Meeting the Paris Agreement on climate change requires substantial investments in low-emissions energy and significant improvements in end-use energy efficiency. These measures can also deliver improved air quality and there is broad recognition of the health benefits of decarbonising energy. Monetising these health benefits is an important part of a robust assessment of the costs and benefits of renewable energy and energy efficiency programs (clean energy programs (CEP)) and a variety of methods have been used to estimate health benefits at national, regional, continental and global scales. Approaches, such as unit damage cost estimates and impact pathways, differ in complexity and spatial coverage and can deliver different estimates for air pollution costs/benefits. To date, the monetised health benefits of CEP in Australia have applied international and global estimates that can range from 2–229USD/tCO2 (USD 2016). Here, we calculate the current health damage costs of coal-fired power in New South Wales (NSW), Australia's most populous state, and the health benefits of CEP. Focusing on PM2.5 pollution, we estimate the current health impacts of coal-fired power at 3.20USD/MWh, approximately 10% of the generation costs, and much lower than previous estimates. We demonstrate the need for locally specific assessment of the air pollution benefits of CEP and illustrate that without locally specific information, the relative costs/benefits of CEP may be significantly over- or understated. We estimate that, for NSW, the health benefits from CEP are 1.80USD/MWh and that the current air pollution health costs of coal-fired power in NSW represent a significant unpriced externality.

**Keywords:** clean energy; energy efficiency; air pollution; health benefits

## 1. Introduction

Limiting global warming to well below 2 °C requires substantial changes to the way that we generate and use electricity [1–4]. Decarbonising electricity supply calls for investments in low-emissions generation, whether through variable renewable sources such as wind, wave and solar or dispatchable renewables from hydroelectricity or biomass combustion [4]. Carbon capture and storage of emissions from fossil fuel generation can also contribute to decarbonisation, and bio-energy carbon capture and storage is likely to be a viable contribution in achieving negative emissions [1]. These measures also act to reduce toxic air pollutants either directly, or through reducing the emission of pollutant precursors [5].

There is extensive literature showing that least cost decarbonisation of the energy sector is likely to include significant improvements in end-use energy efficiency [6–8]. Improving residential, commercial and industrial energy efficiency has been a focus of many

national and sub-national emissions reduction programs [9–11]. Commercial building and residential energy consumption from heating, ventilation and air conditioning, cooking, lighting and entertainment, consume 40% of primary energy globally [12]. More efficient industrial processes and manufacturing can also contribute significantly to decarbonising the electricity supply [13–15].

At the same time, air pollution continues to be a significant environmental concern globally, with estimates of annual premature deaths from fine particles (PM2.5) of 4.2 million persons per year [16]. Air pollution and fossil fuel energy use are closely linked and there is broad recognition of the health benefits of reducing fossil fuel energy emissions [5,17,18].

Many actions that reduce emissions from coal-fired power stations, whether direct PM2.5 emissions or gaseous precursors of secondary aerosols, are likely to improve air quality and deliver improved population health outcomes [17–20]. These avoided health costs are a benefit of CEP and are likely to be material to cost-benefit analysis [21].

A range of approaches have been used to monetise the estimated health benefits from CEP [20,22–24]. These vary from simple "damage cost" approaches—where changes in air pollution damages are valued by changes in pollutants from specific source locations [25]—to impact pathway assessment where chemical transport models (CTM) are used to track emissions and chemical transformations in the atmosphere from emission source through to population exposure and associated health impacts [26].

These approaches have been applied in many regions of varying size with energy systems of varying complexity including: global assessments based on coarse resolution CTM and coarse emissions of global emissions reduction from energy and other sources [22]; regional assessments in Europe [21] using emissions from the Global Energy Assessment, China using global energy demand modelling and local multi-scale emissions estimates coupled with a CTM [23], and the USA using regional emissions models, CTM and impact assessment focussed on benefits from energy efficiency measures on summertime air quality [24]. Where these impacts are monetised, their values vary from global and regional estimates of 31USD/MWh (USA, [24]), 43USD/MWh (Chile, [27]), 16–44EUR/MWh (Greece, [28]), 28USD/MWh (South Korea, [29]).

In Australia, to date there have been few full impact pathway assessments that price the impact of power station emissions and the benefits of CEP on human health. Instead, research on Australian electricity costs either monetises human health impacts from electricity generation based on health damage costs from overseas studies [30], uses simplified damage cost apportionment based on monitoring data [31] or uses solely primary generation emissions that do not take account of secondary aerosol formation [32]. In some studies, the monetised human health costs are ignored even though other climate change externalities are explicitly priced into economy-wide modelling [33]. Our objective is to provide additional estimates of monetized impact/benefit of coal-fired power/CEP.

In this study, we aim to quantify the current health burden of coal-fired power station emissions on the Greater Metropolitan Region of New South Wales, Australia (GMR). We then seek to estimate the likely health benefits from reducing coal-fired power emissions through a suite of clean energy programs. Finally, we monetise these population health costs/benefits with an aim to inform formal economic cost-benefit analysis for CEP type programs in the region.

We begin by assessing current population weighted PM2.5 exposure using CTM. We then model the variation in electricity generation from coal-fired power stations occurring due to lower electricity demand driven by a suite of CEP. We then use the CTM to model changes in PM2.5 related just to the reduction in emissions from power generation driven by the CEP. Finally, we estimate and monetise the current health burden from coal-fired power stations and the future health benefits of CEP, using a health impact assessment. We conclude with observations of the applicability of the work to other regions in Australia and elsewhere.

## 2. Materials and Methods

### 2.1. Local Context

The state of NSW is approximately 809,000 km$^2$ occupying 10% of the Australian continent. It is the most populous state in the nation (8.2 million people) and at AUD 422 billion per annum (FY18) is the largest contributor to national gross-domestic product.

The NSW Government has set a target of achieving net-zero emissions by 2050 and has funded programs forecast to deliver a 50% reduction on 2005 emissions by 2030. Annual greenhouse gas emissions for NSW are 136.6 Tg CO2e, 17% lower than emissions in 2005 [34]. Emissions are dominated by fossil fuel use for electricity generation (51 Tg CO2e) and transport (28 Tg CO2e) [34]. Electricity generation in NSW is largely black coal, which delivers 79% of total dispatched electricity [34]. Promoting clean energy is therefore a priority of the NSW Government and a core part of achieving net zero emissions.

Outside of major events such as bushfires and continental scale dust storms, concentrations of common air pollutants are lower in NSW than many parts of the world, with annual average PM2.5 levels across the state usually below the World Health Organization (WHO) guideline for PM2.5 [35]. However, since typical annual average PM2.5 is above the theoretical minimum risk exposure level [24], PM2.5 pollution continues to have a significant impact on human health and the state's economy [34,36]. The health cost of particle pollution in the Greater Sydney region is estimated to be around AUD 6.6 billion (2016 AUD) per year [37]. A considerable fraction of the monetised benefits is attributable to avoided cases of PM-related deaths. In Sydney, 420–430 premature deaths (about 2% of total deaths) can be attributable to PM2.5 exposure and substantial health benefits are attainable with even modest reductions in PM2.5 air pollution [36,38].

The focus of this study is the Greater Metropolitan Region (GMR) of NSW. The region covers the three largest cities of Sydney (4.9 m), Newcastle (0.5 m) and Wollongong (0.3 m) with a total population across the region of approximately 5.6 million people. The region includes significant coal deposits and six large coal-fired power stations (Bayswater, Eraring, Liddell, Mt Piper, Vales Point, Eraring) were operating during 2013 (the base year of this study). The region is also covered by an extensive air quality monitoring network (Figure 1) [39].

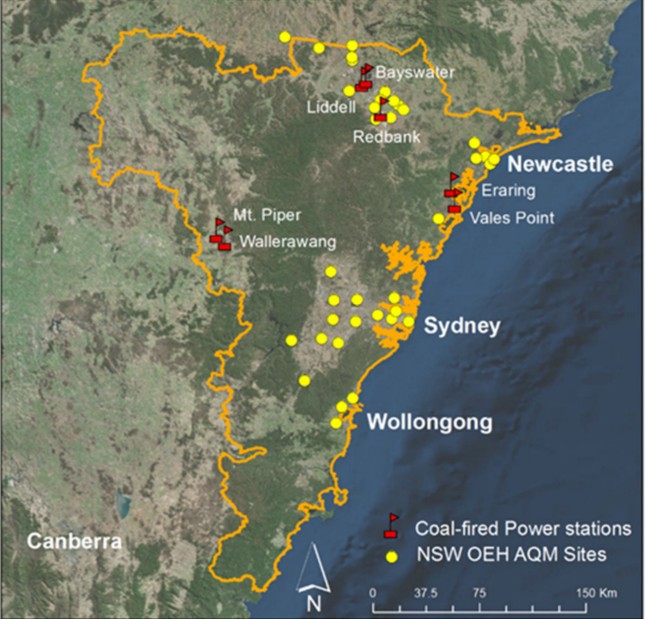

**Figure 1.** Location of air quality monitoring sites and coal-fired power stations in operation in 2013.

## 2.2. Electricity Generation Modelling

We modelled energy generation outlook for the Australian National Electricity Market (NEM) referencing the scenarios and assumptions used by the Australian Energy Market Operator (AEMO) for long term planning purposes [40]. Business as usual (baseline) energy generation from individual coal-fired power stations in NSW to 2044–2045 and two 'demand shock' scenarios reflecting potential changes to generation due to energy efficiency and clean energy are modelled.

The implications of demand reduction in the medium demand shock scenario were modelled based on NSW Government energy efficiency and generation projections for potential energy measures [41]. The measures include improved product standards; energy use information disclosure programs; professional support for identifying energy efficiency opportunities in small and medium businesses; support for small scale renewable generation; regulatory and planning support for large scale renewable generation. Potential energy efficiency measures were projected to save 1379 GWh per year in 2020 and 4028 GWh per year in 2025, with additional renewable energy measures projected to add 1259 GWh in wind and solar generation in 2020. Renewable energy was split equally between wind and solar PV, with average NSW wind power and rooftop solar PV generation shapes used.

For the large demand shock scenario, the reduction in demand due to the CEP measures was assumed to be double the medium demand shock scenario, but to stay constant after the last reported year.

The modelling reflects market announcement to retire the Liddell power station (2051 MW) in the 2022–2023 financial year and Bayswater (2640 MW) in 2035–2036, significant withdrawals from the 10,291 MW of coal generation currently in the NSW fleet. The modelling also considers plant retirements in other states as these may affect power generation in NSW.

## 2.3. Chemical Transport Modelling

PM2.5 from power stations is either emitted directly as primary particles or is formed as secondary atmospheric aerosols from gaseous chemical precursors. Hourly emissions from coal-fired power stations were calculated based on reported/projected energy generation rates for historic/future years using power station-specific emission factors. Emission rates were estimated for primary particles and for several secondary aerosol precursors including sulphur dioxide, sulphur trioxide/sulphuric acid and nitrogen oxides. Reductions in emissions due to the medium and large demand shocks were quantified based on the changes in energy generation rates from the projected base case. Other human and natural sources of air pollution were also modelled to accurately simulate atmospheric chemistry and account for cumulative air pollution.

Regional meteorological and chemical transport modelling was undertaken using the conformal cubic atmospheric model (CCAM) and chemical transport model (CTM) (CCAM-CTM) modelling system [42]. The CTM accounts for both primary and secondary particle pollution. CCAM is a semi-implicit, semi-Lagrangian atmospheric climate model based on the conformal cubic grid [43]. CCAM was driven by the European Reanalysis Interim (ERA-Interim) data to downscale into four domains of 80 km × 80 km, 27 km × 27 km, 9 km × 9 km and 3 km × 3 km resolution and with 35-vertical levels (Figure 2).

The CTM is a three-dimensional Eulerian chemical transport model capturing emissions, transport, chemical transformations and wet and dry deposition within a coupled gas and aerosol phase atmospheric system. Gas and particulate-phase species at the model boundaries are adapted from a global run of the United Kingdom Chemistry and Aerosol scheme for the UK Met Office Unified Model [44]. Photochemistry was modelled using an extended version of the Carbon Bond 5 [45] with updated toluene chemistry [46]. Secondary inorganic aerosols were assumed to exist in thermodynamic equilibrium with gas phase precursors and were modelled using the ISORROPIA-II model [47]. Secondary organic aerosol was modelled using the Volatility Basis Set approach [48].

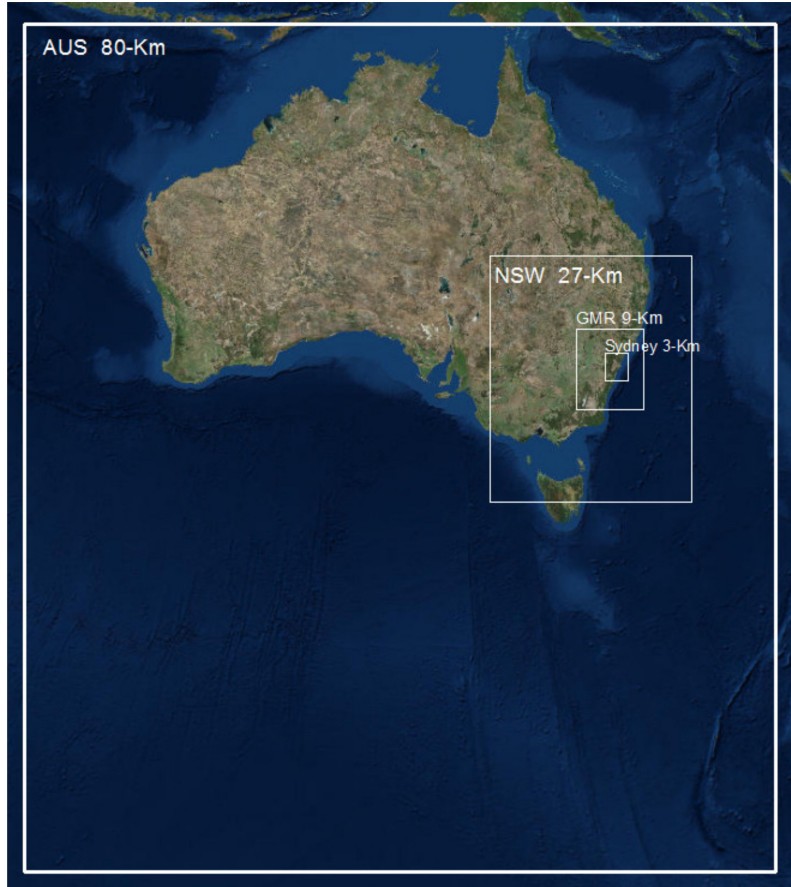

**Figure 2.** CCAM-CTM nested modelling domains.

Detailed particle size and component information is required as input into the CTM to simulate particle nucleation, coagulation and condensational growth. Due to emissions inventory data constraints, the CTM was run in single moment model with particle size and chemical speciation information provided for PM2.5 and PM10. CCAM-CTM in our configuration has previously been validated for regional air quality in the NSW GMR [35,42].

*2.4. Emissions Modelling*

Emissions were projected for a base case year (2013) to support model performance, and for future years assuming business as usual and medium and large demand shock scenarios. We used source and emissions data from the NSW Government Air Emissions Inventory (https://www.epa.nsw.gov.au/your-environment/air/air-emissions-inventory (accessed on 22/10/2021)). Coal-fired power generation emissions accounted for 85% of $SO_2$ emissions and 46% of NOx emissions within the NSW Greater Metropolitan Region (Table 1).

Hourly emissions were calculated for each coal-fired boiler stack based on power station specific emission factors and historical reported or projected future energy generation (see energy generation modelling above). Emissions from other anthropogenic sources including on-road vehicles, non-road diesel and marine sources, industrial emissions and commercial and domestic sources were taken from the inventory. Emissions from continental-scale, natural sources including wind-blown dust, biogenic emissions and marine aerosol are primarily influenced by prevailing meteorology and were calculated in-line within the CTM.

**Table 1.** Source contributions to NSW Greater Metropolitan Region emissions (2013, kilotons per annum).

| Source Group | Oxides of Nitrogen | PM2.5 | Sulfur Dioxide |
|---|---|---|---|
| Coal-fired power generation | 139 | 1 | 198 |
| Industrial and other power generation | 22 | 15 | 16 |
| On-Road Mobile | 45 | 2 | 0 |
| Off-Road Mobile | 59 | 3 | 11 |
| Domestic-Commercial | 4 | 8 | 0 |
| Natural | 36 | 77 | 8 |
| Total | 305 | 106 | 233 |

Rather than modelling air quality for all future years in the time series, we split the data into three periods with similar trends of total emissions/emission reductions and contributions from individual power stations: (i) 2018–2022, (ii) 2023–2034 when Liddell Power Station is retired but Bayswater Power Station is still in operation; and (iii) 2034–2044 after the retirement of Bayswater. We then selected one year within each of these periods to be representative of the changes within the periods (2020, 2028, 2039) (Figure 3).

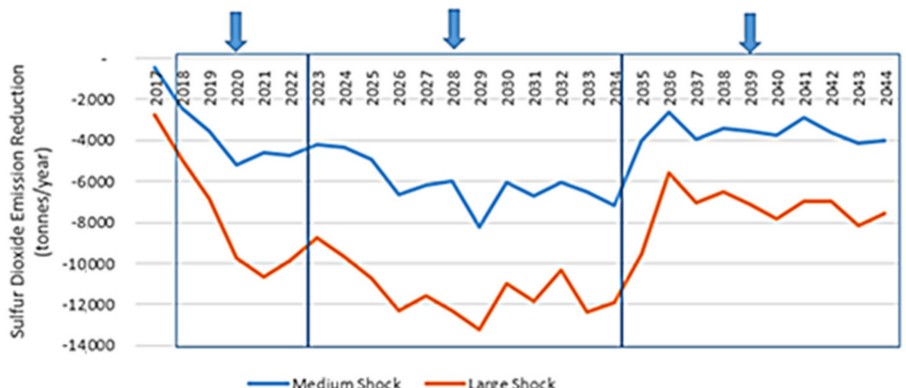

**Figure 3.** Reduction in SO$_2$ emissions from NSW coal-fired power generation projected due to demand shocks showing years selected for regional air quality modelling.

Our focus was on changes in PM2.5 due only to changes in coal-fired power station emissions, hence we held meteorology and non-power station emissions unchanged for all years. We chose to model meteorology for 2013 for all scenarios as we focus on health impacts due to changes in emissions.

*2.5. Model Performance Assessment*

Reference was made to model bias, error and correlation benchmarks for regional airshed modelling used to inform air policy, as documented in the literature [49]. Modelled meteorology and air quality for the base case year were found to be within a reasonable range when compared to measurements from NSW government air quality monitoring stations and Australian Bureau of Meteorology automatic weather stations [50–53].

Modelled hourly SO$_2$ concentrations were within a factor of two of observations, with mean fractional bias MFB) within ±60% for all months and stations and within ±30% in some cases, and the Index of Agreement (IOA) was in the range of 0.51 to 0.77 across stations. The root mean square error (RMSE) was within the observed standard deviation of hourly averages. In the case of PM2.5, model performance is reasonable with all monitoring stations within the acceptable model performance criteria for MFB (±60%) and mean fractional error, MFE (±75%).

### 2.6. Population Exposure Modelling

Population-weighted annual average PM2.5 (pwaa-PM2.5) [54] concentrations were calculated for the NSW greater metropolitan area based on the modelled PM2.5 levels and Australian Bureau of Statistics Statistical Area 2 (SA2) usual residential population data from the 2011 census.

The pwaa-PM2.5 attributable to 2013 coal-fired boiler stack emissions (0.15 $\mu$g/m$^3$) was used to estimate the health burden and related costs associated with long-term exposure to PM2.5 from coal-fired power stations (both primary and secondary) (Figure 4). Changes to pwaa-PM2.5 concentrations due to changes in coal-fired boiler stack emissions given the medium and large demand shock scenarios were estimated relative to the base case with reductions of 4% and 8% projected, respectively.

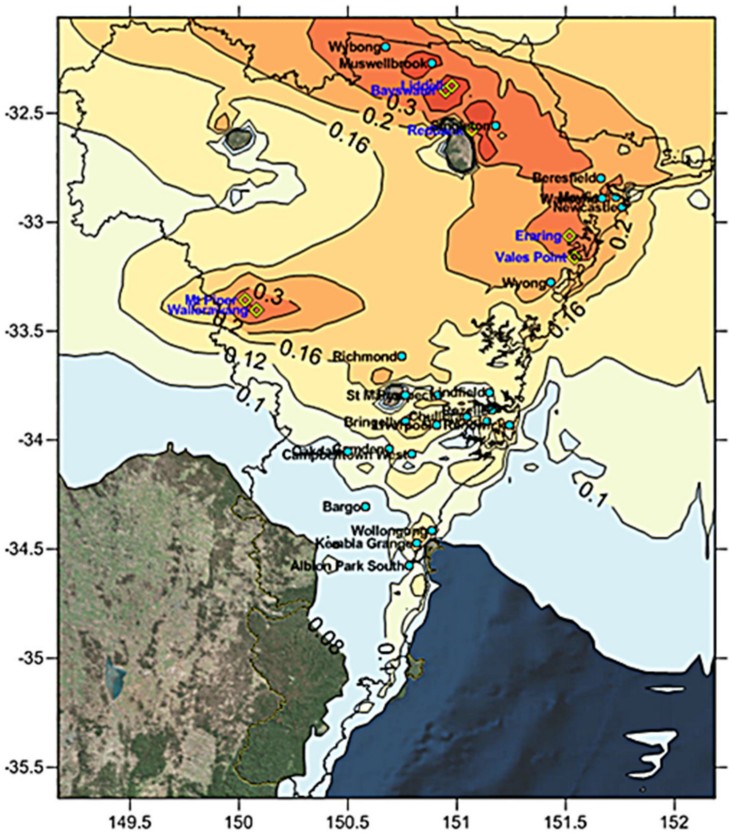

**Figure 4.** Predicted annual average PM2.5 concentrations ($\mu$g m$^{-3}$) due to 2013 coal-fired power station boiler stack emissions.

### 2.7. Health Impacts and Cost Assessments

The health impact assessment covered the NSW greater metropolitan area which covers 85% of the total NSW population including the three largest cities of Sydney, Newcastle and Wollongong and focused on the health burden attributable to long-term exposure to PM2.5 concentrations and the potential health benefits due to reductions in the pwaa-PM2.5 from CEP. The health burden was quantified using attributable numbers of premature deaths (AN), loss of life expectancy at birth and years of life lost (YLL). Health benefits and related costs were quantified based on the estimated number of life years (LY) gained. We conducted two types of sensitivity analysis. Sensitivity analysis determines how different values of an independent variable affect a particular dependent variable under a given set of assumptions. It is very common in different domains. With the health impact estimates we examined the variation in health impacts using two different concentration response functions, two different emissions scenarios and two different lag-

effects. Finally, we tested the sensitivity of the economic valuations using three different discount rates (discount rates reflect the long-term cost of capital, see below).

Health impact estimates are based on a concentration-response coefficient relating the pwaa-PM2.5 to a risk of death. Meta-analysis across large multi-city studies indicates that effects of PM2.5 on health are similar across geographic areas. A concentration-response coefficient derived from a meta-analysis of North American and European cohort studies was used for the central estimate; concentration-response coefficient, β-coefficient, of 0.006 (95% confidence interval 0.004–0.008) [55]. This β-coefficient has previously been applied for assessing health impacts in the Greater Sydney region [36,38]. Sensitivity analysis was conducted using a higher β-coefficient of 0.0121 derived from the reanalysis of the 6-Cities study [56].

Age-specific mortality and population data from the NSW Ministry of Health were applied in the study, with health burden quantified for all-cause mortality in adult populations (aged 30+ years). A delay or "cessation lag" was assumed for the central estimate, i.e., a lag in the time from when the change in the pwaa-PM2.5 levels occur to when health benefits are fully realised. The 20-year distributed lag structure published by the US Environmental Protection Agency [56] was applied, with sensitivity testing undertaken for a 'no cessation lag' case to assess how this affected the results [57].

We used economic valuation methods to assess the potential economic impacts of changes in PM2.5 based on AN, YLL and LY gained. Valuations are sensitive to assumptions regarding the monetary value of statistical life (*VSL*) and discount rates. We applied willingness-to-pay estimates for *VSL* of AUD 7.85 million (2016 AUD, based on ASCC 2008) for central estimates with sensitivity analysis of AUD 4.37 and AUD 10.6 million (2016 AUD, based on OBPR 2014) and used these to estimate the value of a statistical life year (*VSLY*) and applied these to the YLL and LY gained estimates from the health assessment. The value of a statistical life year (*VSLY*) is:

$$VSLY = \frac{r(VSL)}{\left[1 - (1+r)^{-L}\right]} \tag{1}$$

where *r* is the discount rate, to account for the health effects occurring over some period, and *L* is the recommended lifespan of 40 years [58]. *VSLY* is a common approach to valuing mortality impacts from air pollution and other hazards [59,60].

Discount rates are designed to reflect the long-term cost of capital and potential opportunity costs for that capital. There are disagreements about the most appropriate application of discount rates for economic assessment of climate policy, both empirical and conceptual [61]. We apply a discount rate of 7% (sensitivity 3%, 10%) to VSL and VSLY estimates (Table 2) consistent with NSW government guidelines and reflective of the long-term cost of market capital. Consequently, our results reflect more finance-equivalent rather than social-welfare-equivalent estimates [62]. Additionally, we added an annual inflation factor of 2% to account for the expected increase in the willingness of people to pay to reduce health risks as people's incomes rise over time [61].

**Table 2.** Values for VSL and VSLY used in this study. Values for central estimates are in bold. Values are AUD 2016.

| VSL | VSLY, Discounted | | |
|---|---|---|---|
| | 3% | 7% | 10% |
| 4,370,000 | 189,000 | 327,000 | 446,000 |
| **7,850,000** | 340,000 | **589,000** | 803,000 |
| 10,600,000 | 458,000 | 795,000 | 1,084,000 |

## 3. Results

We began by quantifying the health burden due to long-term exposure to current coal-fired power station emissions, focusing on all-cause premature mortality from long-term exposure to PM2.5 as this has been found to be the most health endpoint responsible for the majority of health costs, i.e., prior analyses suggest that the value of avoided premature deaths accounts for over 95% of the total value associated with mortality and morbidity endpoints [53]. We estimated emissions from the calendar year 2013 using the NSW air emissions inventory and modelled the transport and transformation of these emissions using a chemical transport model (CTM). We estimated the contribution of power station emissions to the overall exposure of the greater Sydney population to PM2.5 during 2013 and the associated mortality burden using two values of the PM2.5 and mortality concentration response function ($\beta$ = 0.003, 0.0131). We found that the 57,379 GWh of coal power generated in NSW in 2013 could be attributed to 31 (68) premature deaths and 382 (832) years of life lost with monetised costs of 3.35USD/MWh (USD 1.87–9.84) and 3.05USD/MWh (1.70–8.97 USD), respectively (Table 3).

**Table 3.** Mortality burden of coal-fired power generation in the NSW Greater Metropolitan Region emissions (2013).

| β | Premature Deaths | | Years of Life Lost | |
|---|---|---|---|---|
| | AN | USD/MWh | YLL | USD/MWh |
| 0.0060 | 31 | 3.35 (1.87–4.52) | 382 | 3.05 (0.98–5.62) |
| 0.0131 | 68 | 7.29 (4.05–9.84) | 832 | 6.65 (2.13–12.24) |

To assess the health benefits of reducing PM2.5 air pollution due to power stations in NSW we modelled a future package of energy efficiency and renewable energy programs that together work to reduce overall electricity demand and consequently, emissions from coal plants. Together we estimate these actions to deliver annually up to 8000 GWh of energy savings through efficiency measures and 2500 GWh from new renewable generation (additional to baseline modelled increases in renewable generation).

The CEP measures impact differently across individual power stations. Generators with higher marginal costs, such as Eraring and Vales Point, tend to have larger reductions in their energy output, compared to lower cost plants. After the scheduled retirement of aging plants (Liddell 2022, Vales Point 2027, Bayswater 2035), the remaining stations are projected to increase their energy output. The retirement of older power stations makes the remaining coal generators less likely to be marginal on the merit order, i.e., less likely to have their generation reduced by CEP driven reductions in demand, resulting in stable generation rates.

Health benefits due to reductions in long-term exposure to PM2.5 resulting from CEP measures were quantified based on estimated life years (LY) gained. Health benefits were estimated for four scenarios comprising medium and large demand shocks for two periods and covering the entire modelled period until extinction of the population cohort (Figure 5).

Based on the central estimate assumptions, the medium and large demand shocks were estimated to result in 448 and 922 life years gained, respectively, with a resulting monetary value of AUD 78 million and AUD 159 million (AUD 2016), respectively, for the 2017–2118 time horizon, and 431 and 889 life years gained, with a resulting monetary value of AUD 70 million and AUD 144 million (AUD 2016) for the 2026–2118 time horizon, (which excludes program ramp-up period).

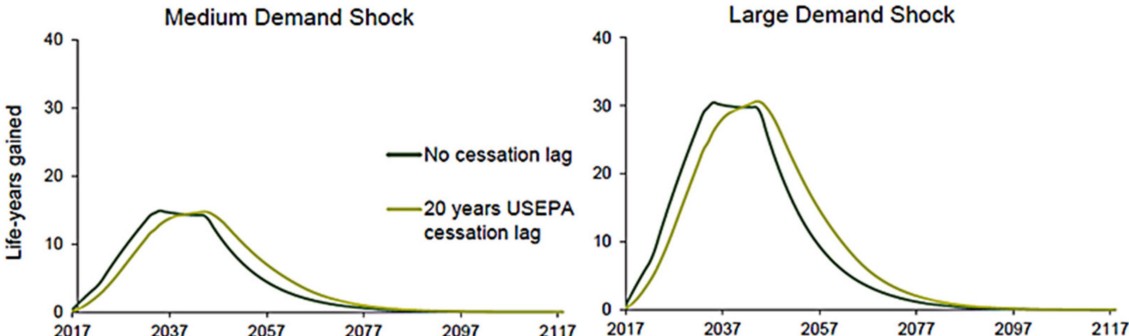

**Figure 5.** Central estimates of LY gained for the extended GMR population over the 2017–2118 period for medium and large demand shock scenarios, without lag and with the USEPA distributed lag.

Based on life years gained for the medium demand shock scenario and the 2026–2118 period (excluding ramp up), and assuming a 7% discount rate, damage costs were estimated to be 2.40AUD/MWh total energy generation (central estimate, Table 4). These damage costs are higher than the central damage costs of 1.50AUD/MWh total energy generation calculated for 2017–2118 and including the measure ramp up period.

**Table 4.** Health benefit of CEP measures in the NSW Greater Metropolitan Region emissions. AUD2016/MWh using 7% discount rate.

| Scenario | Damage Costs (AUD/MWh) | | |
|---|---|---|---|
| | Lower | Central | Upper |
| Medium demand shock (2017–2118, incl. ramp up) | 0.83 | 1.50 | 4.42 |
| Large demand shock (2017–2118, incl. ramp up) | 0.85 | 1.52 | 4.49 |
| Medium demand shock (2026–2118, excl. ramp up) | 1.33 | 2.40 | 7.06 |
| Large demand shock (2026–2118, excl. ramp up) | 1.36 | 2.45 | 7.23 |

Two-time horizons are considered when calculating health benefits in Table 4. The first involves estimating health benefits over the full 2017–2118 period, including the initial period when energy measures are being implemented and ramped up (2017–2025). This time horizon allows the total number of life years gained to be estimated; however, to estimate health benefits in terms of damage costs (AUD/MWh), benefits need to be calculated for the period when the energy measures are fully implemented, i.e., 2026–2118 which excludes the ramp up period. Total LY gained estimates are higher when health benefits gained during the ramp up period (2017–2025) are included, with health benefits calculated over the 2017–2118 period. About 4% of the total life years gained over the 2017–2118 period is estimated to accrue because of the ramp up period, hence the damage costs including the ramp-up period are lower than those excluding the ramp up.

## 4. Discussion

There are three major issues in using international damage cost estimates for Australian assessments, related to the locations and operation of coal-fired power stations.

First, unlike Europe, Asia and North America, Australia is sparsely populated with average occupancy of 3.1 persons/km$^2$. The population density of NSW is 9.7 persons/km$^2$ and even the largest city, Sydney, has a population density significantly less than many large European and Asian cities (Paris, London, Madrid, Tokyo, Seoul) but comparable to many large North American cities (Toronto, San Francisco, Miami). The lower population densities of Australian cities limit the applicability of damage cost estimates derived from regions where population density, and hence potential spatial unit exposure to pollution, is higher. Applying these estimates to Australian cities will likely overstate the health impacts and costs of coal-fired power on human health.

Secondly, from the 1960s onwards many large Australia cities closed in-city power stations and relocated coal-fired generation nearer to coal fields, away from major population centres [63]. For example, nearly all major black and brown coal generators in Australia are more than 100 km from a major city. The relative remoteness of the power stations results in significant dispersion of power station emissions before pollution plumes reach major cities. However, this remoteness also allows time for chemical transformation of gaseous precursors into secondary aerosols within the plume [64].

Finally, bituminous coal used for electricity generation in Australia is of high quality, with low ash, moisture and sulphur contents [65], leading to lower sulphur dioxide emissions from black coal generation compared to plants elsewhere with similar technology using coal with higher sulphur and ash contents.

Together these differences between Australian and other black coal-fired generators limits the applicability of European, Asian or North American damage cost estimates to Australian health impact assessments. Robust estimates of health damage costs in Australia therefore require full impact pathway assessment. These assessments combine emissions estimates from electricity generators with other anthropogenic and natural emissions sources and couple these with a chemical transport model. This allows the assessment of the impacts of both the primary emissions from individual generators and their contribution to secondary aerosols on pollution exposure across the population.

## 5. Conclusions

To date there have been few estimates of the health benefits of CEP in Australia and Australian researchers and government agencies have typically relied on international estimates as inputs to cost-benefit analysis (CBA). We used a full impact pathway assessment to estimate the current damage costs of coal-fired power in NSW and the likely health benefits from a suite of CEP programs. Our results deliver lower estimates of the current impacts of coal-fired power on population health than similar work in North America, Europe and China, demonstrating that robust economic analysis of CEP requires regionally specific information. Without this information the relative costs and benefits of CEP programs may be significantly over- or understated.

Regardless of how our results compare with international estimates, we demonstrate that for NSW there is significant health benefits from CEP programs ranging from 1–5AUD/MWh. Our estimates of current health costs of coal-fired power in NSW are approximately 10–15% of the total costs of coal-fired electricity generation, representing a significant externality that is not currently factored into electricity prices.

Although robust, this study provides a conservative estimate of damages, since it does not assess health impacts and related costs associated with ambient concentrations of other air pollutants such as ozone, sulphur dioxide ($SO_2$) and nitrogen dioxide ($NO_2$) (although the contributions of $SO_2$ and $NO_2$ emissions to secondary sulphate and nitrate particles is accounted for). Similarly, other impacts associated with air pollution that can be monetised, such as impacts on morbidity and other health endpoints, ecosystems, climate and visibility, are not considered. Individually, these additional impacts are likely to be immaterial to the outcomes of CBA for CEP. However, collectively these impacts may be enough to warrant closer consideration within CBAs. The inclusion of these additional impacts into CBAs should be considered on a regionally specific basis.

While ours is a regionally constrained study, its findings have broader applicability. Firstly, we demonstrate the need for locally-specific information if local/sub-national entities (including individual utilities) wish to fully cost the impacts/benefits of their electricity systems/assets. Even though our damage costs appear low compared to international studies, at 10–15% of generation costs, they are nevertheless material for CBA.

We show the application of a system that considers regionally-specific context, uses full impact pathway methods and applies locally relevant emissions, exposure, health response and valuation data. This approach is applicable to any regionally defined system where data are available.

Finally, we demonstrate that in an integrated electricity network with dispersed generation assets and multiple generation technologies, the present damage costs of generation are not equivalent to the future benefits of clean energy programs on an AUD/MWh-to-AUD/MWh basis. Policy makers hence need to ensure that when estimating the expected emissions reduction from existing generation assets through CEP, benefits are assessed through full system impact modelling, and not by application of the current damage cost functions.

**Author Contributions:** Conceptualization, M.M., Y.S. and M.L.R.; methodology, M.M., Y.S., R.A.B., G.G.M. and B.J.; validation, M.M. and Y.S.; formal analysis, M.M., Y.S., R.A.B., G.G.M. and B.J.; resources, M.L.R.; writing—original draft preparation, M.L.R.; writing—review and editing, M.M., Y.S., R.A.B., G.G.M., B.J. and M.L.R. All authors have read and agreed to the published version of the manuscript.

**Funding:** This research received no external funding.

**Institutional Review Board Statement:** Not applicable.

**Informed Consent Statement:** Not applicable.

**Data Availability Statement:** Data from the study are available from the authors upon reasonable request.

**Conflicts of Interest:** The authors declare no conflict of interest.

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
