# Peer review of "Monetising Air Pollution Benefits of Clean Energy Requires Locally Specific Information"

_energies, doi:10.3390/en14227622_

Round 1

Reviewer 1 Report

The paper does not present results that contribute significantly to research on pollution from thermal energy sources. The impacts on air pollution caused by coal-burning energy sources are well known and published. As the research is restricted to a region of Australia, and considering government initiatives and the use of clean sources of energy generation, the results are not applicable globally, which restricts its area of interest.

Author Response

Thank you for your review. 

We acknowledge that our study is focussed on a specific region of Australia, however that does not unduly limit its wider applicability.

In this study we quantify the current health burden of coal-fired power station emissions on the Greater Metropolitan Region of New South Wales, Australia (GMR). We then estimate the likely health benefits from reducing coal fired power emissions through a suite of clean energy programs. Finally, we monetise these population health costs/benefits with an aim to inform formal economic cost-benefit analysis for CEP type programs in the region.

We agree that while the human health impacts of coal fired thermal generation are well known, the scale and estimated economic costs vary significantly from region to region. There are very few studies in Australia that quantify these costs using impact pathway assessments coupled to economic estimators.

Hence we

1) add to the scant literature in Australia on the monetised health impacts from coal-fired power generation and the health benefits of clean energy programs

2) demonstrate that locally specific information is necessary if monetised health impacts of coal-fired power stations, or monetised health benefits of clean energy programs are to be included in traditional cost-benefit analysis

3) quantify the un-costed externality of human health impacts of PM2.5 from coal-fired power generation in this region

4) show that this externality is material, even though our monetised impact costs are are lower than estimates use in Australia based on international estimates.

Together, these results should add to the literature regarding this issue in Australia but also provide guidance for researchers/policy makers in other regions who are seeking to price externalities into their local/regional energy systems.

Further, we believe that our paper meets the aims of energies and adds to the Topical Collection Feature Papers in Energy, Environment and Well-Being.

Reviewer 2 Report

ID: energies-1452943-peer-review-v1

Title: Monetizing air pollution benefits of clean energy requires locally specific information

The authors’ focus is on the calculation of the health damage costs of coal fired power in New South Wales (NSW) state in Australia and the health benefits of CEP. The results show that the current health impacts of coal fired power is much lower than previous estimates and this can be assumed as an achievement in this paper. Authors have concluded that the current air pollution health costs of coal fired power in NSW represent a significant unpriced externality.

In general, the topic can be interest of energy researchers. I have enjoyed reviewing this paper. However, I have some comments about content and organization of the paper. Please see the following comments:

- There are some terms in the paper with no rigid definition. An example is “sensitivity analysis” with the following definition, which is never defined: “Sensitivity analysis determines how different values of an independent variable affect a particular dependent variable under a given set of assumptions. It is very common in different domains.”

- Figure 4 in one of core figures in this paper. The difference between health benefits of four scenarios (in two periods) is not well explained. Can authors please say further what are interesting observations about the difference(s)?

- First, authors should give a summary of literature review (e.g., at the end of Section 1) in order to show their own contributions.

Then, a paragraph should be added to the end of Section 1. As a roadmap, it should have one or two sentences about each of Sections 2-5 to briefly explain the content of each section. It is a standard format of paper writing to help readers to have a quick overview of the paper.

- Equations should be numbered. For instance, the equation of VLSY in Page 8 requires numbering.

- Table 2 is not addressed in the body of the paper. Please address it immediately before (or after) of its appearance in the paper.

- I wonder what software is used to edit the paper. MsWord or Latex? The issue is that the reference list in Pages 11 and 12 does not show the title of papers. I assume it is an issue about compiling the software. Please correct in in the next revision.

-The following papers about CO2 emissions reduction in Australia should be also cited as the scope and domain of research have some similarities with the current paper and can attract many general energy readers. [a] Assessment of contribution of Australia's energy production to CO2 emissions and environmental degradation using statistical dynamic approach. Science of The Total Environment, vol.639, pp. 888-899 [b] The impact of various carbon reduction policies on green flowshop scheduling, Applied Energy, vol.249, pp. 300-315

-Other errors:

Page 9: population cohort (fig. 4) --> population cohort (Fig. 4)

Author Response

We thank the reviewer for their comments which are very helpful. We acknowledge that these comments have improved the paper. Our responses to the suggestions are below.

  1. There are some terms in the paper with no rigid definition. An example is “sensitivity analysis” with the following definition, which is never defined: “Sensitivity analysis determines how different values of an independent variable affect a particular dependent variable under a given set of assumptions. It is very common in different domains.” Accepted. We have added definitions and hope that now there are no uncommon terms that remain undefined.
  2. Figure 4 in one of core figures in this paper. The difference between health benefits of four scenarios (in two periods) is not well explained. Can authors please say further what are interesting observations about the difference(s)? Accepted. We have added text to the paragraph above the table that expands on our results. Thank you for this suggestion which has added value to our results. 
  3. First, authors should give a summary of literature review (e.g., at the end of Section 1) in order to show their own contributions. Then, a paragraph should be added to the end of Section 1. As a roadmap, it should have one or two sentences about each of Sections 2-5 to briefly explain the content of each section. It is a standard format of paper writing to help readers to have a quick overview of the paperAccepted. We have added additional paragraphs at the end of Section 1. This makes the paper introduction clearer and we thank you for this suggestion.
  4. Equations should be numbered. For instance, the equation of VLSY in Page 8 requires numbering. Accepted. Equation now numbered.
  5. Table 2 is not addressed in the body of the paper. Please address it immediately before (or after) of its appearance in the paper. Accepted. Thank you for bringing this oversight to our attention. We have added reference to Table 2 immediately above the table.
  6. I wonder what software is used to edit the paper. MsWord or Latex? The issue is that the reference list in Pages 11 and 12 does not show the title of papers. I assume it is an issue about compiling the software. Please correct in in the next revision. Accepted. We apologise, this was our misinterpretation of the preferred format for references in energies. We have updated now to include article titles.
  7. The following papers about CO2 emissions reduction in Australia should be also cited as the scope and domain of research have some similarities with the current paper and can attract many general energy readers. [a] Assessment of contribution of Australia's energy production to CO2 emissions and environmental degradation using statistical dynamic approach. Science of The Total Environment, vol.639, pp. 888-899 [b] The impact of various carbon reduction policies on green flowshop scheduling, Applied Energy, vol.249, pp. 300-315. Accepted. Thank you for bringing these to our attention. We have included first reference in our expanded Section 1 but feel the second is too specific for this paper.
  8. Other errors: Page 9: population cohort (fig. 4) --> population cohort (Fig. 4). Corrected.

Reviewer 3 Report

                                               Review Report

                                                         on

Monetizing air pollution benefits of clean energy requires locally

                                               (Energies-1452943)

This paper raises the need for specific information to be able to calculate the economic value of the impact of clean energy programs.  

Area of strength

The subject of this paper  is current, relevant and suitable for the Journal.

Areas of weakness

The introduction includes a brief review of the literature on the topic. It would be advisable to separate both sections and delve deeper into each one of them. At no time is the objective of the work specified and the research questions it tries to answer.

Besides, the contribution of the paper should be justified in the introduction, since many of the data used have been extracted from previous studies

The conclusions section should be developed further by pointing out the main contributions of the paper and the practical implications that can be derived from it.

Author Response

We thank the reviewer for their helpful comments and suggestions. We have addressed these and acknowledge that they have improved our paper. Specific actions to address comments follow:

  1. The introduction includes a brief review of the literature on the topic. It would be advisable to separate both sections and delve deeper into each one of them. At no time is the objective of the work specified and the research questions it tries to answer. Accepted. Thank you for this advice. We have added additional paragraphs to the Introduction, including an expanded literature review. We also more clearly state the aim of the study. Together (and with point 2 below) these have added clarity to the section and we thank you for your suggestion. Note that we have not included sub-section headings in the Introduction as we believe that with the additional text under your suggestion, the section now presents more clearly. 
  2. Besides, the contribution of the paper should be justified in the introduction, since many of the data used have been extracted from previous studies. Accepted. See our comments for 1) 
  3. The conclusions section should be developed further by pointing out the main contributions of the paper and the practical implications that can be derived from it. Accepted. We have added additional text to the Conclusion which sharpens our results and better articulates their usefulness and relevance for policy makers. Thank you for this suggestion as it improves our paper markedly.

Round 2

Reviewer 1 Report

The authors carried out an extensive review of the paper, presenting a new version that is able to be published in its current form.

Reviewer 2 Report

I have look the paper once again, mostly with focus on the revised paragraphs. 

Authors could resolve some issues I have mentioned. The format of the paper is better although it is not a well-written paper yet.

I can accept the paper as it is. 

Reviewer 3 Report

All the suggestions have been followed by the authors and I believe that the paper has improved substantially so that it could be published.